# FAdam: Adam is a natural gradient optimizer using diagonal empirical Fisher information

## Abstract

This paper establishes a mathematical foundation for the Adam optimizer, eluci­dating its connection to natural gradient descent through Riemannian and infor­mation geometry. We provide an accessible and detailed analysis of the diagonal empirical Fisher information matrix (FIM) in Adam, clarifying all approximations to bridge the gap between the natural gradient and Adam, and advocating for the use of log probability functions as loss, which should be based on discrete distri­butions, due to the limitations of empirical FIM. Our analysis uncovers flaws in the original Adam algorithm, leading to proposed corrections such as enhanced mo­mentum calculations, adjusted bias corrections, and gradient clipping. We refine the weight decay term based on our theoretical framework. Our modified algo­rithm, **Fisher Adam (FAdam)**, demonstrates superior performance across diverse domains including LLM, ASR, and VQ-VAE, achieving SoTA results in ASR.

## 1 Introduction

Natural Gradient Descent (NGD) is a powerful optimization method that has shown promise in training machine learning models. Introduced by Amari (1998), and revisited recently Pascanu & Bengio (2013), NGD offers an alternative to traditional gradient descent by taking into account the curvature of the loss landscape. This is achieved through the use of the Fisher information matrix (FIM), which provides a Riemannian metric for the statistical manifold. Despite its potential benefits, NGD faces a significant computational challenge: the calculation of FIM, which can be prohibitively expensive for large models like deep neural networks.

Adam (Kingma & Ba, 2014) is the de facto standard optimizer, favored for its fast convergence and practicality. While the original paper mentions the second momentum term can be interpreted as the diagonal FIM, a comprehensive theoretical understanding of the square root FIM, akin to the approach first seen in AdaGrad (Lydia & Francis, 2019), has remained elusive Lin et al. (2024). This is in contrast to NGD, which utilizes the inverse FIM.

**Our main contribution** is to provide an accessible mathematical explanation of the Adam opti­mizer, drawing upon fundamental concepts from Riemannian geometry and information geometry, thereby demonstrating that Adam is an approximation of NGD. This framework enables us to illu­minate the origin of the square root term in Adam and to advocate for the use of a log probability function as the loss function when employing Adam.

We demonstrate that Adam utilizes the diagonal empirical Fisher information, providing a detailed explanation of both the diagonal approximation and the empirical approximation. Notably, the em­pirical approximation suggests that the loss function should be based on discrete distributions, which implies using categorical cross-entropy loss rather than L2 loss in the image domain. Furthermore, our analysis uncovers flaws in the original Adam, leading us to propose corrections such as aver­aging natural gradient by momentum, adjusting bias corrections, and introducing gradient clipping. We also enhance the weight decay using principles of Riemannian geometry. Our modified algo­rithm, named **Fisher Adam (FAdam)**, demonstrates strong performance across various domains, as evidenced by our experiments with Large Language Models (LLMs) for text, Automatic Speech Recognition (ASR) for speech, and Vector Quantized Variational Autoencoder (VQ-VAE) for image. In particular, our ASR experiments achieve new state-of-the-art results.

## 2 BACKGROUND

We use the notation from differential geometry, as introduced in Appendix A.1.

### 2.1 GRADIENT ON RIEMANNIAN MANIFOLD

The inner product in the tangent space of a Riemannian manifold is defined by the Riemannian metric tensor $\mathbf{g} = g_{ij}d\theta^i \otimes d\theta^j$. The Riemannian metric tensor components, $g_{ij}$, are symmetric and positive definite. Therefore, $g_{ij}$ is always invertible, and its inverse is denoted as $g^{ij}$.

$$\vec{v} \cdot \vec{u} := \mathbf{g}(\vec{v}, \vec{u}) = g_{ij}d\theta^i \otimes d\theta^j(v^k\frac{\partial}{\partial\theta^k}, u^r\frac{\partial}{\partial\theta^r}) = g_{ij}v^k u^r d\theta^i(\frac{\partial}{\partial\theta^k}) \otimes d\theta^j(\frac{\partial}{\partial\theta^r}) = g_{ij}v^i u^j \quad (1)$$

The differential form of a scalar field $\phi$ (e.g., loss $\mathcal{L}(\boldsymbol{\theta})$) is defined in the covector space, as shown in Eq. (33). However, since the model parameter $\boldsymbol{\theta}$ resides in the vector space, we need to transform the covector $d\phi$ into the vector space to perform operations involving both $d\phi$ and $\boldsymbol{\theta}$ (e.g., optimizing $\boldsymbol{\theta}$ as shown in Eq. (13)).

The object transformed into the vector space is defined as the gradient $\phi$, denoted as $\nabla\phi$, and its definition is based on the inner product being equivalent to the directional derivative (Eq. (35)), as shown in Eq. (2). By rearranging the terms with respect to $\nabla\phi$, we obtain the following equation for the gradient $\nabla\phi$ in the tangent space $\mathcal{T}_{\boldsymbol{\theta}}\mathcal{M}$, as shown in Eq. (3).[1]

$$(\nabla\phi)^i v^j g_{ij} = \nabla\phi \cdot \vec{v} := d\phi(\vec{v}) = v^j\frac{\partial\phi}{\partial\theta^j} \quad (2)$$

$$(\nabla\phi)^i = g^{ij}\frac{\partial\phi}{\partial\theta^j} \rightarrow \nabla\phi = g^{ij}\frac{\partial\phi}{\partial\theta^j}\frac{\partial}{\partial\theta^i} \quad (3)$$

In Euclidean space, the Riemannian metric $g_{ij}$ is Kronecker delta, so the gradient is usually expressed as follows:

$$\nabla\phi = \frac{\partial\phi}{\partial\theta^i}\frac{\partial}{\partial\theta^i} \quad (4)$$

In ML community, Eq. (4) is often denoted as the **gradient** $\nabla\phi$, while Eq. (3) is denoted as the **natural gradient** $\tilde{\nabla}\phi$. This notation conflicts with conventions in differential geometry. Since our paper is focused on ML, we will adhere to ML conventions from this point forward.

Typically, the component part of tensor operations are represented using matrix multiplication. The metric tensor component, $g_{ij}$, is represented by a matrix $\boldsymbol{G}$, and $\nabla\phi$ is a column vector.

$$\tilde{\nabla}\phi = \boldsymbol{G}^{-1}\nabla\phi \quad (5)$$

Equation 1 can also be expressed using matrix multiplication.

$$\vec{v} \cdot \vec{u} = \boldsymbol{v}^\top \boldsymbol{G}\boldsymbol{u} \quad (6)$$

### 2.2 FISHER INFORMATION MATRIX (FIM) AND NATURAL GRADIENT

The Fisher information quantifies the amount of information an observable random variable x (representing data) conveys about an unknown parameter $\boldsymbol{\theta}$ (representing a model parameter) influencing its probability. Given the probability mass function $P(\mathrm{x}|\boldsymbol{\theta})$ (or probability density function $p(\mathrm{x}|\boldsymbol{\theta})$) for the random variable x, the Fisher information is defined as the variance of the score function (i.e., the gradient of the log-likelihood), which is symmetric and positive semi-definite by definition.

---

[1]$g^{ij}$ is the inverse tensor component of $g_{ij}$, such that $g_{ij}g^{ij} = 1$.

$$\boldsymbol{F}(\boldsymbol{\theta}) := \mathbb{E}_{x \sim P_{\boldsymbol{\theta}}}[\nabla_{\boldsymbol{\theta}} \log P(\mathrm{x}|\boldsymbol{\theta}) \, \nabla_{\boldsymbol{\theta}} \log P(\mathrm{x}|\boldsymbol{\theta})^{\top}]. \tag{7}$$

Fisher information is the expected value of the Hessian matrix. (Proof provided in Appendix A.2.) It represents the curvature of the log-likelihood on the statistical manifold where the model parameters $\boldsymbol{\theta}$ reside.

$$\boldsymbol{F}(\boldsymbol{\theta}) = -\mathbb{E}_{x \sim P_{\boldsymbol{\theta}}}[\nabla_{\boldsymbol{\theta}}^2 \log P(\mathrm{x}|\boldsymbol{\theta})] = -\mathbb{E}_{x \sim P_{\boldsymbol{\theta}}}[\boldsymbol{H}_{\boldsymbol{\theta}}(\log P(\mathrm{x}|\boldsymbol{\theta}))]. \tag{8}$$

In the realm of statistical manifolds, the distance between $\boldsymbol{\theta}$ and $\boldsymbol{\theta} + \boldsymbol{d}$ is quantified by the Kullback-Leibler divergence $D_{\mathrm{KL}}(P(\mathrm{x}|\boldsymbol{\theta})\|P(\mathrm{x}|\boldsymbol{\theta} + \boldsymbol{d}))$. For an infinitesimal displacement $\boldsymbol{d}$, a second-order Taylor series approximation reveals the Fisher information as the underlying distance metric (Amari, 1998; Pascanu & Bengio, 2013; Kristiadi, 2018). A detailed proof can be found in Appendix A.3.

$$D_{\mathrm{KL}}(P(\mathrm{x}|\boldsymbol{\theta})\|P(\mathrm{x}|\boldsymbol{\theta} + \boldsymbol{d})) \approx \frac{1}{2}\boldsymbol{d}^{\top}\boldsymbol{F}(\boldsymbol{\theta})\boldsymbol{d}. \tag{9}$$

As mentioned in Eq. (6), the magnitude of a vector $\boldsymbol{v}$ in the tangent space $\mathcal{T}_{\boldsymbol{\theta}}\mathcal{M}$ can be expressed using the inner product, and the Fisher Information Matrix (FIM) serves as the Riemannian metric Amari (2012).[2] Prior works Berman et al. (2023b); Berman & Klinger (2024); Berman et al. (2023a) have demonstrated the importance of FIM in providing an inherent scale for learning.

$$\vec{v} \cdot \vec{v} = \|\boldsymbol{v}\|^2 = \boldsymbol{v}^{\top}\boldsymbol{G}\boldsymbol{v} = \boldsymbol{v}^{\top}\boldsymbol{F}(\boldsymbol{\theta})\boldsymbol{v} \tag{10}$$

In statistical manifolds, the gradient is described using the Fisher information $\boldsymbol{F}$ in Eq. (5), and this is called the **natural gradient**.

$$\tilde{\nabla}\phi = \boldsymbol{F}^{-1}\nabla\phi \tag{11}$$

The intuitive interpretation of Eq. (11) is that components with higher information undergo conservative movement, while components with lower information exhibit wider movement. [3]

## 3 TOWARD FADAM

### 3.1 LOSS AND FIM

To utilize the natural gradient Eq. (11), we need to know both the gradient term and FIM. The gradient term requires the calculation of the loss $\mathcal{L}(\boldsymbol{\theta})$, while the FIM requires the score function $\nabla_{\boldsymbol{\theta}} \log P(\mathrm{x}|\boldsymbol{\theta})$. If the loss is expressed in the form of $-\log P(\mathrm{x}|\boldsymbol{\theta})$, we eliminate the need to calculate the score function separately. Therefore, for using natural gradient optimizers like Adam[4] Kingma & Ba (2014), **the loss function must be in the form of the log-likelihood**. We will delve deeper into the choice of loss function in Section 3.3.1.

$$\tilde{\nabla}\mathcal{L}(\boldsymbol{\theta}) = \boldsymbol{F}^{-1}\nabla\mathcal{L}(\boldsymbol{\theta}) = \mathbb{E}_{x \sim P_{\boldsymbol{\theta}}}[\nabla_{\boldsymbol{\theta}} \log P(\mathrm{x}|\boldsymbol{\theta}) \, \nabla_{\boldsymbol{\theta}} \log P(\mathrm{x}|\boldsymbol{\theta})^{\top}]^{-1}\nabla_{\boldsymbol{\theta}} - \log P(\mathrm{x}|\boldsymbol{\theta}) \tag{12}$$

The model parameter $\boldsymbol{\theta}$ is updated using the given natural gradient $\tilde{\nabla}\mathcal{L}(\boldsymbol{\theta})$, where $\eta$ is the learning rate.

---

[2]Since FIM is symmetric and positive semi-definite, it produces a Pseudo-Riemannian manifold Amari (2012).

[3]The inverse FIM exhibits a close relationship with the covariance matrix of the log likelihood, denoted as $\Sigma_{\boldsymbol{\theta}}^{-1}$. The Mahalanobis distance, $d_M(\boldsymbol{x}, \mathcal{P})^2$, which measures the distance between a data point $\boldsymbol{x}$ and a distribution $\mathcal{P}$, is defined as $d_M(\boldsymbol{x}, \mathcal{P})^2 := (\boldsymbol{x} - \boldsymbol{\mu})^{\top}\Sigma_{\boldsymbol{x}}^{-1}(\boldsymbol{x} - \boldsymbol{\mu})$ (Mahalanobis, 2018). Comparing this to Eq. (10), we discern a connection between the inverse Fisher information and the covariance matrix: higher covariance indicates lower information. This parallel echoes the principle in information theory where higher entropy corresponds to lower information.

[4]We will discuss why Adam is considered a natural gradient optimizer in Section 3.4.

$$\boldsymbol{\theta}_{t+1} = \boldsymbol{\theta}_t - \eta \boldsymbol{F}^{-1} \nabla \mathcal{L}(\boldsymbol{\theta}) \tag{13}$$

Natural gradient is considered a second-order method because FIM is the expected value of the Hessian, as shown in Eq. (8). A comparison with Newton's method is provided in Appendix B.1.

## 3.2 DIAGONAL FISHER INFORMATION

One key reason for Adam Kingma & Ba (2014)'s significant success compared to other second-order methods is its memory complexity. Adam scales linearly with the number of parameters, $O(N)$, while methods using FIM typically scale quadratically, $O(N^2)$. For models with billions of parameters, $O(N^2)$ is impractical.

Adam utilizes the diagonal Fisher information matrix. As discussed in 2.2, the inverse FIM approximates the covariance of the log likelihood. The diagonal FIM results in the loss of all covariance information except for the variances. Interestingly, Adam's success story suggests that the loss of covariance information might not be detrimental in practice.

Let $\hat{\boldsymbol{f}}(\boldsymbol{\theta})$ denote the diagonal FIM [5] obtained by diagonalizing FIM in Eq. (7). The gradient of loss function (12) is greatly simplified.

$$\hat{\boldsymbol{f}}(\boldsymbol{\theta}) := \mathbb{E}_{\mathrm{x} \sim P_{\boldsymbol{\theta}}}[\nabla_{\boldsymbol{\theta}} \log P(\mathrm{x}|\boldsymbol{\theta})^2] \tag{14}$$

$$\tilde{\nabla}\mathcal{L}(\boldsymbol{\theta}) = \hat{\boldsymbol{f}}^{-1}\nabla\mathcal{L}(\boldsymbol{\theta}) = -\frac{\nabla_{\boldsymbol{\theta}}\log P(\mathrm{x}|\boldsymbol{\theta})}{\mathbb{E}_{\mathrm{x} \sim P_{\boldsymbol{\theta}}}[\nabla_{\boldsymbol{\theta}}\log P(\mathrm{x}|\boldsymbol{\theta})^2]} \tag{15}$$

Amari et al. (2019) proves that the off-diagonal elements of FIM are smaller than the diagonal elements by an order of $1/\sqrt{n}$, where $n$ represents the number of elements in the matrix. This finding justifies the use of the quasi-diagonal natural gradient method when the weight matrices of each layer are sufficiently large like LLM (Large Language Models).

Meanwhile, there have been efforts to capture important off-diagonal elements. For example, approximation methods utilizing low-rank approximations Roux et al. (2007); Mu et al. (2022) and Kronecker-factored approximations Martens & Grosse (2015); Gupta et al. (2018); Anil et al. (2019); Martins Gomes et al. (2024) have been explored. The application of off-diagonal FIM to Adam is left for future study.

## 3.3 EMPIRICAL FISHER INFORMATION

To compute the diagonal FIM in Eq. (14), the expected value needs to be calculated with respect to the parametric probabilistic model $P(\mathrm{x}|\boldsymbol{\theta})$. While data can be sampled from the parametric model, it is not always a straightforward process. Various sampling methods exist, such as Gibbs sampling, Langevin Markov chain Monte Carlo (MCMC) sampling Parisi (1981), Metropolis-Hastings MC sampling Neal et al. (2011) and the recent GFlowNet (Bengio et al., 2021). However, none of these methods are universally efficient in generating sufficient data with high fidelity and diversity.

Instead of $P(\mathrm{x}|\boldsymbol{\theta})$, we utilize the true data-generating distribution $p_{data}(\mathrm{x})$ to compute FIM. Although the exact form of $p_{data}(\mathrm{x})$ is unknown, we have access to a training set of samples. We compute FIM by utilizing the training set $\mathcal{D}$ by substituting the true distribution $p_{data}(\mathrm{x})$ with the empirical distribution $\hat{p}_{data}(\mathrm{x})$. This approximated FIM is referred to as the **empirical FIM** in the statistics community (Kunstner et al., 2019). The training set might not contain enough samples of low-probability x, which could lead to issues with the empirical FIM.

$$\hat{\boldsymbol{f}}(\boldsymbol{\theta}) = \mathbb{E}_{\mathrm{x} \sim P_{\boldsymbol{\theta}}}[\nabla_{\boldsymbol{\theta}} \log P(\mathrm{x}|\boldsymbol{\theta})^2] \approx \mathbb{E}_{\mathrm{x} \sim p_{data}}[\nabla_{\boldsymbol{\theta}} \log P(\mathrm{x}|\boldsymbol{\theta})^2], \tag{16}$$

$$\approx \mathbb{E}_{\mathrm{x} \sim \hat{p}_{data}}[\nabla_{\boldsymbol{\theta}} \log P(\mathrm{x}|\boldsymbol{\theta})^2] = \frac{1}{|\mathcal{D}|}\sum_{\mathrm{x} \in \mathcal{D}} \nabla_{\boldsymbol{\theta}} \log P(\mathrm{x}|\boldsymbol{\theta})^2 \tag{17}$$

---

[5]Actually, from this point forward, the diagonal FIM is a vector.

The expected value of the loss function is also obtained from the empirical distribution rather than the true distribution. Optimizing this cost function $J(\boldsymbol{\theta})$ is referred to as empirical risk minimization, for similar reasons.

$$J(\boldsymbol{\theta}) = -\mathbb{E}_{\mathrm{x} \sim p_{data}}[\log P(\mathrm{x}|\boldsymbol{\theta})] \approx -\mathbb{E}_{\mathrm{x} \sim \hat{p}_{data}}[\log P(\mathrm{x}|\boldsymbol{\theta})] = -\frac{1}{|\mathcal{D}|}\sum_{\mathrm{x} \in \mathcal{D}} \log P(\mathrm{x}|\boldsymbol{\theta}) \quad (18)$$

$$\nabla_{\boldsymbol{\theta}} J(\boldsymbol{\theta}) = \nabla_{\boldsymbol{\theta}} \mathbb{E}_{\mathrm{x} \in \mathcal{D}}[\mathcal{L}(\boldsymbol{\theta})] = \frac{1}{|\mathcal{D}|}\sum_{\mathrm{x} \in \mathcal{D}} \nabla_{\boldsymbol{\theta}} \mathcal{L}(\boldsymbol{\theta}) = -\frac{1}{|\mathcal{D}|}\sum_{\mathrm{x} \in \mathcal{D}} \nabla_{\boldsymbol{\theta}} \log P(\mathrm{x}|\boldsymbol{\theta}) \quad (19)$$

Calculating the exact cost function and FIM over the entire training set is computationally expensive. Therefore, in practice, the expected value is approximated using a minibatch $\mathcal{B}$. As this approximation further increases the uncertainty in the empirical FIM, FIM is typically estimated using an exponential moving average (EMA). Therefore, during training, natural gradient is computed as follows:

$$\tilde{\nabla} J(\boldsymbol{\theta}) = \hat{\boldsymbol{f}}^{-1} \nabla J(\boldsymbol{\theta}) \approx -\mathbb{E}_{\mathrm{x} \in \mathcal{B}}[\nabla_{\boldsymbol{\theta}} \log P(\mathrm{x}|\boldsymbol{\theta})] / \mathbb{E}_{EMA}[\mathbb{E}_{\mathrm{x} \in \mathcal{B}}[\nabla_{\boldsymbol{\theta}} \log P(\mathrm{x}|\boldsymbol{\theta})^2]] \quad (20)$$

$$\approx -\mathbb{E}_{\mathrm{x} \in \mathcal{B}}[\nabla_{\boldsymbol{\theta}} \log P(\mathrm{x}|\boldsymbol{\theta})] / \mathbb{E}_{EMA}[\mathbb{E}_{\mathrm{x} \in \mathcal{B}}[\nabla_{\boldsymbol{\theta}} \log P(\mathrm{x}|\boldsymbol{\theta})]^2] \quad (21)$$

$$= -\boldsymbol{g} / \mathbb{E}_{EMA}[\boldsymbol{g}^2] \quad (22)$$

Let the gradient of a minibatch be denoted as $\boldsymbol{g}$. To reuse $\boldsymbol{g}$ for calculating FIM, Adam makes another approximation from Eq. (20) to Eq. (21). Wang & Aitchison (2024) showed that using Eq. (20) makes Adam less sensitive to batch size. Adam variants are recommended to use large batch sizes Smith et al. (2017); Kunstner et al. (2023) to accurately estimate not only the gradient but also FIM.

In supervised learning, the loss function becomes conditional log-likelihood, necessitating specific considerations detailed in Appendix B.2.

### 3.3.1 DISCRETE VS CONTINUOUS PROBABILITY DISTRIBUTIONS

It's worth noting that while Adam often outperforms SGD in the text domain, several studies have reported that its convergence point in the image domain, particularly for generative tasks or when using CNNs, can be inferior to that of SGD Luo et al. (2019); Yuan & Gao (2020); Zhang et al. (2020); Jelassi et al. (2021); Kunstner et al. (2023). Empirical evidence indicates that Adam excels when dealing with discrete distributions, such as text inputs with categorical distributions. However, it may encounter difficulties when handling continuous distributions, such as image inputs with Gaussian distributions.

In the image domain, using the L2 loss is common practice due to its equivalence to the negative log-likelihood of a Gaussian distribution under the gradient. $k$ represents the partition function of the Gaussian distribution, which is equal to $1/\sqrt{2\pi\sigma^2}$. Since it is independent of $\boldsymbol{\theta}$, it is eliminated when the gradient is taken.

$$\nabla_{\boldsymbol{\theta}} J(\boldsymbol{\theta}) \approx -\mathbb{E}_{\mathrm{x} \sim \hat{p}_{data}}[\nabla_{\boldsymbol{\theta}} \log p(\mathrm{x}|\boldsymbol{\theta})] = -\mathbb{E}_{\mathrm{x} \sim \hat{p}_{data}}\left[\nabla_{\boldsymbol{\theta}} \log k \exp{-\left(\frac{\mathrm{x}-\boldsymbol{\mu}(\boldsymbol{\theta})}{\sqrt{2}\sigma}\right)^2}\right] \quad (23)$$

$$= \frac{1}{2\sigma^2}\mathbb{E}_{\mathrm{x} \sim \hat{p}_{data}}\left[\nabla_{\boldsymbol{\theta}}(\mathrm{x}-\boldsymbol{\mu}(\boldsymbol{\theta}))^2\right] = \frac{1}{2\sigma^2|\mathcal{D}|}\sum_{\mathrm{x} \in \mathcal{D}} \nabla_{\boldsymbol{\theta}}(\mathrm{x}-\boldsymbol{\mu}(\boldsymbol{\theta}))^2 \quad (24)$$

We performed empirical approximations to calculate the expected value of FIM. Eq. (16) represents the empirical approximation of FIM for generative models, while Eq. (52) and Eq. (58) represent the empirical approximations for discriminative models. We hypothesize that these empirical approximations cause significantly more problems in continuous distributions than in discrete distributions. This disparity arises from the fundamental difference in how expected values are calculated for discrete and continuous distributions. Discrete distributions rely on probability mass functions, where

the softmax function often concentrates the majority of probability mass on a few top logits. This concentration allows for relatively accurate empirical approximations. [6] [7] In contrast, continuous distributions require integration over their probability density functions, making the estimation of their expected values with a single sampled value an overly simplified and potentially inaccurate approximation.

Kunstner et al. (2019) argue against the use of the empirical Fisher in natural gradient descent, providing examples solely with continuous inputs and distributions.[8] Their findings support our assertion that Adam may not perform well with continuous distributions, while the empirical success of Adam suggests that the empirical Fisher is adequate for discrete distributions.

We propose utilizing **log-likelihood loss based on discrete probability distributions**. In the image domain, this translates to using **cross-entropy (CE) loss on a categorical distribution instead of the L2 loss**.[9] This can be implemented by modifying predictions to utilize a one-hot encoding, predicting 256 values per RGB channel. As demonstrated in Section 4.3, this modification significantly enhances the FID (Fréchet Inception Distance) metric for VQ-VAE Van Den Oord et al. (2017).

It has also been reported that categorical CE loss improves accuracy in **floating-point number regression** tasks Imani & White (2018); Schrittwieser et al. (2020); Sønderby et al. (2020); Hafner et al. (2023); Farebrother et al. (2024). Farebrother et al. (2024) reported that CE loss with a histogram-based discretization (HL-Gauss Imani & White (2018)) significantly improved the prediction of rewards and values in reinforcement learning.

The exceptional scalability of Large Language Models (LLMs) (Brown et al., 2020; Kaplan et al., 2020) with model size can likely be attributed to two factors: the inherently discrete nature of text data and the widespread use of the Adam optimizer. As LLMs continue to evolve into foundation models Bommasani et al. (2021) for various modalities, image, speech, and video domains are also adopting discrete token representations Borsos et al. (2023); Team et al. (2023); Lu et al. (2023). This provides further evidence that empirical FIM estimation necessitates the use of discrete distributions.

### 3.4 FISHER ADAM

#### 3.4.1 RECIPROCAL VS RECIPROCAL SQUARE-ROOT FOR FIM

We have discovered that the second momentum term in Adam Kingma & Ba (2014) closely resembles a natural gradient with a diagonal empirical FIM, as demonstrated in Eq. (22) and Eq. (60). However, there is a key distinction: while natural gradients are divided by the FIM, Adam divides gradients by the square root of the FIM, as shown in Algorithm 3.

To precisely update $\theta_t$ at each training step as per Eq. (13), we ideally need an accurate estimation of FIM. However, the empirical FIM computed by the minibatch data $\mathcal{B}$ is noisy. Relying on a diagonal empirical FIM can result in zero components, which causes the natural gradient to diverge due to

---

[6]Label smoothing in classification tasks Szegedy et al. (2016) has been shown to enhance performance. A theoretical explanation is that it results in a more accurate estimation of the empirical FIM. This is because label smoothing leads to a better approximation of the empirical FIM by computing all possible labels (y), rather than relying on a single data sample, as shown in Eq. (57).

[7]Knowledge distillation Hinton (2015) is employed to enhance the performance of Large Language Models Gunter et al. (2024); Team et al. (2024); Dubey et al. (2024). This technique leverages the teacher model's probability distribution to calculate the expected value of the student model's log probability, enabling Adam to estimate FIM more accurately. While knowledge distillation is recognized for its ability to transfer dark knowledge from teacher to student, its effectiveness may also be attributed to the improved optimization facilitated by the precise FIM estimation within Adam.

[8]Regrettably, relying on image classification (i.e., continuous inputs) or, even more restrictively, simple regression like curve fitting (i.e., continuous distribution) to analyze Adam or FIM may limit the generalizability of findings Kunstner et al. (2019); Reddi et al. (2019); Cohen et al. (2022).

[9]Diffusion models Ho et al. (2020) seem to perform well with L2 loss. It is likely because the expectation of the loss is taken over not only image samples but also the noisy diffusion steps, which cover the model distribution sufficiently, allowing for a more accurate empirical estimate of the FIM. This could be a significant factor contributing to the success of diffusion models.

---

**Algorithm 1** Fisher Adam (FAdam)

---
1: **given** $\beta_1 = 0.9$, $\beta_2 = 0.999$, $\epsilon = 10^{-15}$, $c = 1$, $\lambda = 0.001$, $\rho = 0.5$, $\eta_t$
2: **initialize** $\boldsymbol{\theta}_0$, $t \leftarrow 0$, $\boldsymbol{m}_0 \leftarrow 0_N$, $\boldsymbol{f}_0 \leftarrow 1_N$       ▷ FIM init to 1 as per Section 3.4.4
3: **repeat**
4:     $t \leftarrow t + 1$
5:     $\boldsymbol{g}_t \leftarrow \nabla_{\boldsymbol{\theta}} \log P_t(\boldsymbol{\theta}_{-1})$       ▷ Stochastic gradient as per Eq. (12)
6:     $\hat{\beta}_2 \leftarrow \beta_2(1 - \beta_2^{t-1})/(1 - \beta_2^t)$       ▷ Bias correction as per Section 3.4.4
7:     $\boldsymbol{f}_t \leftarrow \hat{\beta}_2 \boldsymbol{f}_{t-1} + (1 - \hat{\beta}_2)\boldsymbol{g}_t^2$       ▷ EMA diagonal empirical FIM as per Section 3.4.1
8:     $\bar{\boldsymbol{g}}_t \leftarrow \boldsymbol{g}_t/(\boldsymbol{f}_t^{\rho} + \epsilon)$       ▷ Invariant natural gradient as per Eq. (27)
9:     $\bar{\boldsymbol{g}}_t \leftarrow \bar{\boldsymbol{g}}_t/\max(1, \mathrm{RMS}(\bar{\boldsymbol{g}}_t)/c)$       ▷ Clip the gradient as per Appendix B.3
10:    $\boldsymbol{m}_t \leftarrow \beta_1 \boldsymbol{m}_{t-1} + (1 - \beta_1)\bar{\boldsymbol{g}}_t$       ▷ EMA momentum as per Section 3.4.2
11:    $\bar{\boldsymbol{g}}_w \leftarrow \boldsymbol{\theta}_{t-1}/(\boldsymbol{f}_t^{\rho} + \epsilon)$       ▷ Weight decay as per Eq. (28)
12:    $\bar{\boldsymbol{g}}_w \leftarrow \bar{\boldsymbol{g}}_w/\max(1, \mathrm{RMS}(\bar{\boldsymbol{g}}_w)/c)$       ▷ Clip weight decay as per Appendix B.3
13:    $\boldsymbol{\theta}_t \leftarrow \boldsymbol{\theta}_{t-1} - \eta_t(\boldsymbol{m}_t + \lambda\bar{\boldsymbol{g}}_w)$       ▷ Update $\boldsymbol{\theta}$ as per Eq. (13)
14: **until** *stopping criterion is met*
15: **return** optimized parameters $\boldsymbol{\theta}_t$

---

division by zero. To address this and obtain a more stable diagonal empirical FIM, an exponential moving average (EMA) is employed.

The gradient and FIM in Eq. (22) vary at each point $\boldsymbol{\theta}$. As shown in Section 2.1, not only their components change but also their underlying basis. However, EMA averages the components of the Fisher information tensor obtained across different $\boldsymbol{\theta}$ points, despite their potentially differing basis. This is a mathematically invalid operation, except for in Euclidean space.[10] As $\beta_2$=0.999 is standard value in Adam, EMA has a half-life of 700 steps [11], meaning it averages FIM over roughly 1000 steps. During this time, the basis can shift significantly as the model parameter moves to a new $\boldsymbol{\theta}$ location on the manifold.

In differential geometry, scalars, vectors and tensors are invariant under a coordinate change. The length of a vector, derived from the inner product (Eq. (1)), is a scalar. For a given gradient, we can express its length as shown in Eq. (10), and simplify it using the diagonal FIM in Eq. (26).

We only have access to the components of the gradient vector, and the invariant quantity is the vector length. However, EMA attempts to average the vector components as if they were in Euclidean space, ignoring the changing basis. To address this, we propose constructing a gradient vector (Eq. (27)) in Euclidean space that has the same length as the known vector length (Eq. (26)), and then apply EMA to this artificial Euclidean vector.

We refer to it as the **invariant natural gradient**, denoted as $\bar{\nabla} J(\boldsymbol{\theta})$. This is **the reason why Adam uses the square root**, as shown in Algorithm 3. In Appendix C.1.1, we conduct an ablation study on the exponent of the FIM term, and find that the square root is the optimal choice.

$$\|\tilde{\nabla} J(\boldsymbol{\theta})\| = \tilde{\nabla} J(\boldsymbol{\theta}) \cdot \tilde{\nabla} J(\boldsymbol{\theta}) = (\boldsymbol{F}^{-1}\nabla J(\boldsymbol{\theta}))^{\top} \boldsymbol{F} (\boldsymbol{F}^{-1}\nabla J(\boldsymbol{\theta})) = \nabla J(\boldsymbol{\theta})^{\top} \boldsymbol{F}^{-1}\nabla J(\boldsymbol{\theta}) \quad (25)$$

$$\approx \nabla J(\boldsymbol{\theta})^{\top} \hat{\boldsymbol{f}}^{-1}\nabla J(\boldsymbol{\theta}) = \|\nabla J(\boldsymbol{\theta})/\sqrt{\hat{\boldsymbol{f}}}\|^2 \quad (26)$$

$$\bar{\nabla} J(\boldsymbol{\theta}) := \nabla J(\boldsymbol{\theta})/\sqrt{\hat{\boldsymbol{f}}} \quad (27)$$

Adam variants like AdaDelta Zeiler (2012), AdaMax Kingma & Ba (2014) and Yogi Zaheer et al. (2018) modify the second momentum to further deviate from the natural gradient. A more principled approach would focus on improving the estimation of the empirical FIM, incorporating off-diagonal elements, and investigating how to make using natural gradient, instead of invariant natural gradient.

---

[10]Properly adding tensors from different tangent spaces requires parallel transport, which relies on understanding how the manifold's basis changes. This is determined by the Riemannian metric tensor field, but accurately knowing even the FIM at a single point is computationally expensive, making knowledge of the entire FIM field intractable.

[11]$\sim \log(0.5)/\log(0.999)$

If we were able to achieve a more accurate FIM estimation with smaller $\beta_2$, it might be possible to eliminate the square root operation.

### 3.4.2 MOMENTUM

Momentum methods Rumelhart et al. (1986) employ exponential moving average (EMA) of gradients along the optimization trajectory to mitigate the noise inherent in stochastic gradients estimated from minibatches $\mathcal{B}$.

At a point $\boldsymbol{\theta}$ on the statistical manifold, the ideal gradient is the invariant natural gradient, as shown in Eq. (27). Therefore, momentum should average the invariant natural gradient $\overline{\nabla} J(\boldsymbol{\theta})$ rather than the raw gradient, as shown in Algorithm 1. In contrast, as shown in Algorithm 3, Adam Kingma & Ba (2014) calculates the first and second momentums separately and then combines them, lacking a clear theoretical justification. In Algorithm 4, Adafactor Shazeer & Stern (2018), on the other hand, already aligns with our proposed approach. LaProp Ziyin et al. (2020) has empirically demonstrated the benefits of this modification. Furthermore, the second momentum is not momentum. We refer to it as FIM.

FAdam omits zero bias correction in EMA of momentum [12], because excessive gradient descent is unnecessary before the momentum is sufficiently accumulated. This approach can be viewed as built-in informed warmup schedule. Adafactor implementation Shazeer (2018) already omits zero bias correction for momentum, although this is not explicitly mentioned in the paper Shazeer & Stern (2018).

### 3.4.3 WEIGHT DECAY ON MANIFOLD

AdamW Loshchilov & Hutter (2017), which decouples the weight decay term from the loss and applies it directly to the Adam optimizer, has demonstrated general performance improvements and is now widely used as a standard practice. Our mathematical framework provides a sound theoretical explanation for this observed phenomenon. The loss function should be log-likelihood, but the weight decay term has nothing to do with the probability distribution. If the weight decay term is included in the loss, it causes problems in estimating the FIM. [13]

Weight decay is a great example of how auxiliary losses should be handled. **If the auxiliary loss is related to the log-likelihood, it can be included in the loss. Otherwise, it should be bypassed by the Adam optimizer, as in the case of weight decay**.

Furthermore, following Eq. (11), weight decay should also be applied as a natural gradient. Since we already have the FIM, we can express the weight decay gradient in a similar way to Eq. (27).

$$\boldsymbol{g}_w = \boldsymbol{\theta}/\sqrt{\hat{\boldsymbol{f}}} \tag{28}$$

Intuitively, components with low Fisher information can be pushed closer to zero without significantly impacting the model performance. Elastic weight consolidation Kirkpatrick et al. (2017) leverages a similar concept, utilizing Fisher information to regularize the change of $\boldsymbol{\theta}$.[14]

It has been reported that for training large-scale models like LLMs, decoupling weight decay from the learning rate helps stabilize training Wortsman et al. (2023). However, since our modifications make the weight decay adaptable to the loss surface, such workarounds might not be necessary. Further research is needed to confirm this hypothesis.

---

[12]$\boldsymbol{m}_t \leftarrow \boldsymbol{m}_t/(1 - \beta_1^t)$

[13]While the weight decay term can be seen as a form of prior in a Bayesian framework, it is often directly added to the loss function (equivalent to the log posterior). The AdamW Loshchilov & Hutter (2017) highlights that this practice can negatively impact performance, suggesting that the commonly used L2 weight decay might not accurately represent the true prior of the model.

[14]EWC can directly utilize FIM from FAdam, eliminating the need to compute it separately.

### 3.4.4 FADAM

Fisher Adam (FAdam), incorporating all the discussed modifications, is presented in Algorithm 1. Fisher Adafactor (FAdafactor) modification of Adafactor is presented in Algorithm 2 in Appendix B.4. [15]

### 3.4.5 CONVERGENCE ANALYSIS

Algorithm 1 does not deviate from the assumptions of the convergence analysis presented in Eq. (15) of the latest Adam convergence proof by Défossez et al. (2020). Therefore, FAdam achieves the following convergence guarantees. Specifically, as the number of iterations N becomes sufficiently large, the expected value of the gradient becomes sufficiently small. The detailed explanation is provided in Appendix B.6.

$$\mathbb{E}\left[\|\nabla F(x_\tau)\|^2\right] \leq O(d\ln(N)/\sqrt{N}) \tag{29}$$

## 4 EXPERIMENT

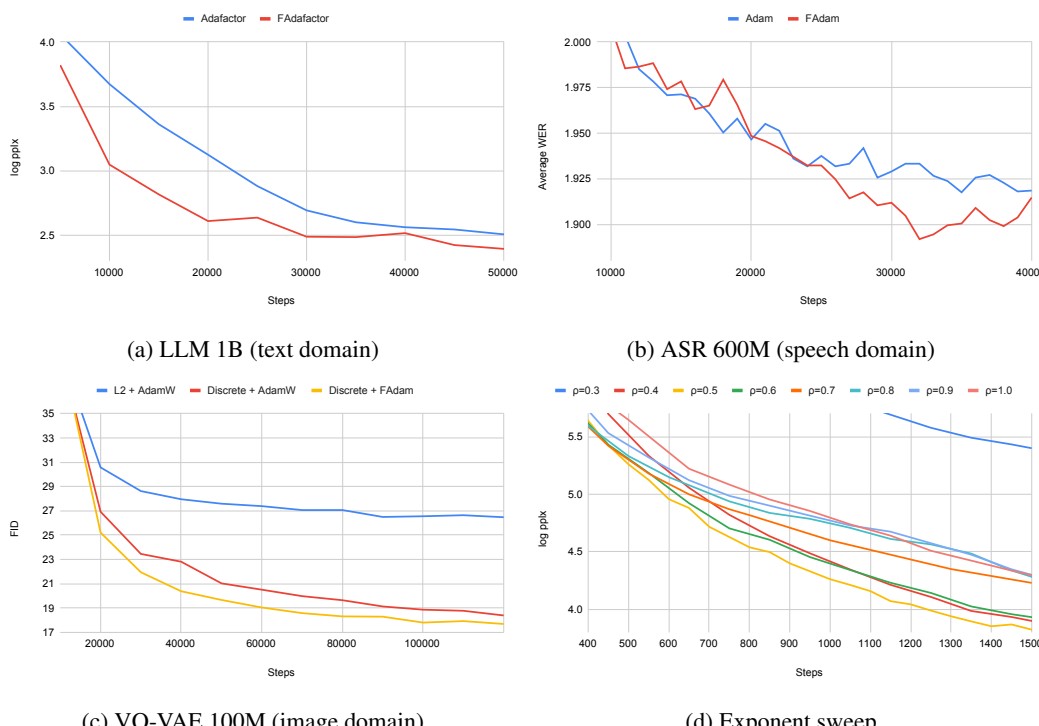

(a) LLM 1B (text domain)

(b) ASR 600M (speech domain)

(c) VQ-VAE 100M (image domain)

(d) Exponent sweep

Figure 1: Comparison of FAdam and Adam performance. (a) Eval loss (log pplx) on 1B LLMs, presenting FAdafactor outperforms Adafactor. (b) Average WER on LibriSpeech using 600M Conformer models, presenting FAdam outperforms Adam. (c) FID of ImageNet generation using 100M VQ-VAE models, presenting FAdam outperforms AdamW. (d) Comparison of FIM exponents on a 1B LLM, showing 0.5 (square root) as the optimal choice.

In our experiments, we took existing state-of-the-art models from various domains and simply replaced their optimizers with FAdam, keeping all other hyperparameters unchanged, except for $\epsilon$.

---

[15]Additionally, we have specified that FIM ($f_0$) is initialized to 1, not 0, in Algorithm 1. This is because FIM represents a Riemannian metric, which defaults to the identity matrix in flat space. However, this change does not affect the logic of the algorithm because bias correction ignores the initial value. The bias correction for FIM is adopted from Adafactor Shazeer & Stern (2018), which is agnostic to the initial value.

Despite this, FAdam consistently outperformed the original Adam variants. Our results demonstrate that FAdam can be effectively used as a drop-in replacement for Adam in real-world applications.

## 4.1 LLM (TEXT DOMAIN)

We pretrained the 1B parameter LLM model from mT5-Large Xue (2020) on mC4 dataset Raffel et al. (2020). FAdafactor and Adafactor shared the same hyperparameters, $\beta_1$=0.9, $\beta_2$=0.99, and $\lambda$=0.001. However, the epsilon value used was $\epsilon$=1e-12 for FAdafactor and $\epsilon$=1e-30 for Adafactor. As demonstrated in Fig. 1a, FAdafactor outperforms Adafactor.

## 4.2 ASR (SPEECH DOMAIN)

The 600M parameter Conformer Gulati et al. (2020) model from the w2v-BERT Chung et al. (2021) is one of the lowest WER (word error rate) achieving models on the LibriSpeech dataset Panayotov et al. (2015). This model was pretrained using w2v-BERT and then finetuned on LibriSpeech data using the RNNT loss Graves (2012). The WER was further improved to SoTA levels using noisy student semi-supervised finetuning Park et al. (2020) on LibriLight Kahn et al. (2020). We compared Adam and FAdam using models that were fine-tuned in a semi-supervised fashion on LibriLight data, paired with semi-supervised pseudo labels.

FAdam and Adam share the same hyperparameters: $\beta_1$=0.9, $\beta_2$=0.98, and $\lambda$=0.001, except for $\epsilon$ (FAdam 1e-12 and Adam 1e-8). As demonstrated in Fig. 1b and Table 1, FAdam not only outperforms Adam but also establishes a new state-of-the-art (SoTA) Word Error Rate (WER) on LibriSpeech for 600M parameter models.

| LibriSpeech WERs | dev | dev-other | test | test-other | avg |
|---|---|---|---|---|---|
| Adam (w2v-BERT paper Chung et al. (2021)) | 1.30 | 2.60 | 1.40 | 2.70 | 2.00 |
| Adam | 1.30 | 2.54 | 1.33 | 2.59 | 1.93 |
| **FAdam** | 1.29 | 2.49 | 1.34 | 2.49 | **1.89** |

Table 1: LibriSpeech WERs

## 4.3 VQ-VAE (IMAGE DOMAIN)

We trained a 100M parameter ViT VQ-GAN model Yu et al. (2021) on the ImageNet dataset Russakovsky et al. (2015). To verify the hypothesis from Section 3.3.1 that categorical cross-entropy (CE) loss is superior to L2 loss on Adam, we exclusively used either CE loss or L2 loss + logit-laplace during training.[16] The VQ-GAN paper Yu et al. (2021) introduced logit-laplace to adjust the output scale when using L2 loss.

FAdam and AdamW Loshchilov & Hutter (2017) share the same hyperparameters: $\beta_1$=0.9, $\beta_2$=0.99, and $\lambda$=1e-4, except for $\epsilon$ (FAdam 1e-15 and AdamW 1e-8). As demonstrated in Fig. 1c, FAdam outperforms AdamW, and categorical CE loss not only outperforms L2 loss but also eliminates the need for the complex logit-laplace transformation.

## 5 CONCLUSION

In this work, we have revealed the mathematical foundation of the Adam optimizer, clarifying the constraints on loss function selection and the approximations involved in its derivation. This groundwork opens avenues for future research to develop improved optimizers by mitigating these approximations. Building upon our theoretical foundation, we have proposed an enhanced algorithm, FAdam, and demonstrated its effectiveness across diverse domains.

---

[16]Since our primary focus is on replacing the L2 loss, we omitted GAN and perceptual losses, making the model effectively a VQ-VAE Van Den Oord et al. (2017) rather than a VQ-GAN.

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

# A  BACKGROUND

## A.1  TENSOR CALCULUS NOTATION

In this paper, we will derive the Adam algorithm from the perspective of information geometry. Since statistical formulas are frequently used in machine learning papers, but Riemannian geometry formulas are not, we will first clarify the notation before proceeding. We follow the notation of Petersen (2006)'s Riemannian geometry textbook.

We define the tangent space at a point $\boldsymbol{\theta}$ on a manifold as $\mathcal{T}_{\boldsymbol{\theta}}\mathcal{M}$. Given a vector $\vec{v} = v^i \boldsymbol{e}_i$ on $\mathcal{T}_{\boldsymbol{\theta}}\mathcal{M}$, the basis vectors are defined as partial derivatives as follows:

$$\boldsymbol{e}_i := \frac{\partial}{\partial \theta^i} \tag{30}$$

Therefore, a vector $\vec{v}$ can be expressed using Einstein notation as follows, with $v^i$ representing the vector components and $\frac{\partial}{\partial \theta^i}$ representing the basis vectors:

$$\vec{v} = \boldsymbol{v} = v^i \frac{\partial}{\partial \theta^i} \tag{31}$$

From an ML practitioner's perspective, the reason for understanding vector is that the parameters $\boldsymbol{\theta}$ of probability density (or mass) functions $p(\boldsymbol{\theta})$ (i.e., scalar fields) reside within a vector space.

Given a covector $\boldsymbol{w} = w_i \boldsymbol{e}^i$ in the dual space of a vector space, the covector basis is defined using a differential one-form.

$$\boldsymbol{e}^i := d\theta^i \tag{32}$$

From an ML practitioner's perspective, the reason for understanding covector is that the differential form of the loss, $d\mathcal{L}(\boldsymbol{\theta})$, reside within a covector space.

$$d\mathcal{L}(\boldsymbol{\theta}) = \frac{\partial \mathcal{L}(\boldsymbol{\theta})}{\partial \theta^i} d\theta^i \tag{33}$$

The operation between the vector basis and the covector basis results in the Kronecker delta. The operation between vector $\vec{v}$ and covector $d\mathcal{L}$ is as shown in Eq. (35). The meaning of Eq. (35) is the directional derivative, representing the derivative of the loss $\mathcal{L}$ in the direction of $\vec{v}$.

$$d\theta^i \left( \frac{\partial}{\partial \theta^j} \right) = \delta^i_j \tag{34}$$

$$d\mathcal{L}(\vec{v}) = \frac{\partial \mathcal{L}(\boldsymbol{\theta})}{\partial \theta^i} v^j d\theta^i \left( \frac{\partial}{\partial \theta^j} \right) = \frac{\partial \mathcal{L}(\boldsymbol{\theta})}{\partial \theta^i} v^i \tag{35}$$

The covariant derivative is an extension of the directional derivative operation from a scalar filed to vector and tensor fields. [17] The covariant derivative of a vector field $\vec{w}$ along a vector $\vec{v}$ is denoted as follows, where $\Gamma^k_{ij}$ represents the Christoffel symbols:

$$D_{\vec{v}}\vec{w} = v^i \frac{\partial}{\partial \theta^i} \left( w^j \frac{\partial}{\partial \theta^j} \right) = v^i \frac{\partial}{\partial \theta^i} w^j \frac{\partial}{\partial \theta^j} + v^i w^j \frac{\partial}{\partial \theta^i} \frac{\partial}{\partial \theta^j} = \left( v^i \frac{\partial}{\partial \theta^i} w^k + v^i w^j \Gamma^k_{ij} \right) \frac{\partial}{\partial \theta^k} \tag{36}$$

---

[17] When applied to a scalar field (i.e., a scalar function), the covariant derivative is called the directional derivative.

## A.2 THE PROOF OF FISHER IS NEGATIVE HESSIAN

The relationship between Fisher information and the negative Hessian's expectation can be proven as follows:

$$\nabla_{\boldsymbol{\theta}}^2 \log P(\mathrm{x}|\theta) = \frac{\nabla_{\boldsymbol{\theta}}^2 P(\mathrm{x}|\theta)}{P(\mathrm{x}|\theta)} - \nabla_{\boldsymbol{\theta}} \log P(\mathrm{x}|\boldsymbol{\theta}) \nabla_{\boldsymbol{\theta}} \log P(\mathrm{x}|\boldsymbol{\theta})^\top \tag{37}$$

$$\mathbb{E}_{\mathrm{x} \sim P_{\boldsymbol{\theta}}} \left[ \frac{\nabla_{\boldsymbol{\theta}}^2 P(\mathrm{x}|\theta)}{P(\mathrm{x}|\theta)} \right] = \int P(\mathrm{x}|\theta) \frac{\nabla_{\boldsymbol{\theta}}^2 P(\mathrm{x}|\theta)}{P(\mathrm{x}|\theta)} d\mathrm{x} = \nabla_{\boldsymbol{\theta}}^2 \int P(\mathrm{x}|\theta) d\mathrm{x} = \nabla_{\boldsymbol{\theta}}^2 1 = 0 \tag{38}$$

$$\mathbb{E}_{\mathrm{x} \sim P_{\boldsymbol{\theta}}} \left[ \nabla_{\boldsymbol{\theta}}^2 \log P(\mathrm{x}|\theta) \right] = -\mathbb{E}_{\mathrm{x} \sim P_{\boldsymbol{\theta}}} \left[ \nabla_{\boldsymbol{\theta}} \log P(\mathrm{x}|\boldsymbol{\theta}) \nabla_{\boldsymbol{\theta}} \log P(\mathrm{x}|\boldsymbol{\theta})^\top \right] \tag{39}$$

## A.3 THE PROOF OF KL APPROXIMATION

As seen in Eq. (9), the Kullback-Leibler (KL) divergence with an infinitesimal displacement $\boldsymbol{d}$ can be approximated by the Fisher information. We provide a proof of this relationship below, following the approach presented in Kristiadi (2018).

$$D_{\mathrm{KL}}(P(\mathrm{x}|\boldsymbol{\theta})\|P(\mathrm{x}|\boldsymbol{\theta}+\boldsymbol{d})) \approx \frac{1}{2}\boldsymbol{d}^\top \boldsymbol{F}(\boldsymbol{\theta})\boldsymbol{d}. \tag{40}$$

By using a second-order Taylor series approximation, the KL divergence can be approximated as follows:

$$D_{\mathrm{KL}}(P_{\boldsymbol{\theta}}\|P_{\boldsymbol{\theta}+\boldsymbol{d}}) \approx D_{\mathrm{KL}}(P_{\boldsymbol{\theta}}\|P_{\boldsymbol{\theta}}) + (\nabla_{\boldsymbol{\theta}'} D_{\mathrm{KL}}(P_{\boldsymbol{\theta}}\|P_{\boldsymbol{\theta}'})|_{\boldsymbol{\theta}=\boldsymbol{\theta}'})^\top \boldsymbol{d} + \frac{1}{2}\boldsymbol{d}^\top \nabla_{\boldsymbol{\theta}'}^2 D_{\mathrm{KL}}(P_{\boldsymbol{\theta}}\|P_{\boldsymbol{\theta}'})\boldsymbol{d} \tag{41}$$

$D_{\mathrm{KL}}(P_{\boldsymbol{\theta}}\|P_{\boldsymbol{\theta}})$ is 0 by the definition of KL divergence. The first-order approximation term also becomes 0 through the following process.

$$\nabla_{\boldsymbol{\theta}'} D_{\mathrm{KL}}(P_{\boldsymbol{\theta}}\|P_{\boldsymbol{\theta}'}) = \underbrace{\nabla_{\boldsymbol{\theta}'} \mathbb{E}_{\mathrm{x} \sim P_{\boldsymbol{\theta}}}[\log P_{\boldsymbol{\theta}}]}_{} - \nabla_{\boldsymbol{\theta}'} \mathbb{E}_{\mathrm{x} \sim P_{\boldsymbol{\theta}}}[\log P_{\boldsymbol{\theta}'}] = -\mathbb{E}_{\mathrm{x} \sim P_{\boldsymbol{\theta}}}[\nabla_{\boldsymbol{\theta}'} \log P_{\boldsymbol{\theta}'}] \tag{42}$$

$$= -\int P_{\boldsymbol{\theta}} \nabla_{\boldsymbol{\theta}'} \log P_{\boldsymbol{\theta}'}|_{\boldsymbol{\theta}=\boldsymbol{\theta}'} d\mathrm{x} = -\int P_{\boldsymbol{\theta}} \frac{\nabla_{\boldsymbol{\theta}'} P_{\boldsymbol{\theta}'}}{P_{\boldsymbol{\theta}'}}|_{\boldsymbol{\theta}=\boldsymbol{\theta}'} d\mathrm{x} \tag{43}$$

$$= -\int \nabla_{\boldsymbol{\theta}} P_{\boldsymbol{\theta}} d\mathrm{x} = -\nabla_{\boldsymbol{\theta}} \int P_{\boldsymbol{\theta}} d\mathrm{x} = -\nabla_{\boldsymbol{\theta}} 1 = 0 \tag{44}$$

The Fisher information emerges from the second-order approximation term, as shown by utilizing the Hessian property in Eq. (8).

$$\nabla_{\boldsymbol{\theta}'}^2 D_{\mathrm{KL}}(P_{\boldsymbol{\theta}}\|P_{\boldsymbol{\theta}'}) = \underbrace{\nabla_{\boldsymbol{\theta}'}^2 \mathbb{E}_{\mathrm{x} \sim P_{\boldsymbol{\theta}}}[\log P_{\boldsymbol{\theta}}]}_{} - \nabla_{\boldsymbol{\theta}'}^2 \mathbb{E}_{\mathrm{x} \sim P_{\boldsymbol{\theta}}}[\log P_{\boldsymbol{\theta}'}] \tag{45}$$

$$= -\mathbb{E}_{\mathrm{x} \sim P_{\boldsymbol{\theta}}}\left[\nabla_{\boldsymbol{\theta}'}^2 \log P_{\boldsymbol{\theta}'}|_{\boldsymbol{\theta}=\boldsymbol{\theta}'}\right] = -\mathbb{E}_{\mathrm{x} \sim P_{\boldsymbol{\theta}}}\left[\nabla_{\boldsymbol{\theta}}^2 \log P_{\boldsymbol{\theta}}\right] \tag{46}$$

$$= \boldsymbol{F}(\boldsymbol{\theta}) \tag{47}$$

Therefore, equation Eq. (9) is proven.

# B  TOWARD FADAM

## B.1  COMPARED TO NEWTON'S SECOND-ORDER METHOD

As shown in Eq. (8), FIM is equivalent to the Hessian of the loss.

$$\boldsymbol{F}(\boldsymbol{\theta}) = -\mathbb{E}_{\mathrm{x} \sim P_{\boldsymbol{\theta}}}[\boldsymbol{H}_{\boldsymbol{\theta}}(\log P(\mathrm{x}|\theta))] = \mathbb{E}_{\mathrm{x} \sim P_{\boldsymbol{\theta}}}[\boldsymbol{H}_{\boldsymbol{\theta}}(\mathcal{L}(\boldsymbol{\theta}))]. \tag{48}$$

Therefore, $\boldsymbol{\theta}$ update Eq. (13) incorporates the Hessian term.

$$\boldsymbol{\theta}_{t+1} = \boldsymbol{\theta}_t - \eta \mathbb{E}_{\mathrm{x} \sim P_{\boldsymbol{\theta}}}[\boldsymbol{H}_{\boldsymbol{\theta}}(\mathcal{L}(\boldsymbol{\theta}))]^{-1} \nabla \mathcal{L}(\boldsymbol{\theta}) \tag{49}$$

As FIM is the expected value of the Hessian, natural gradient optimization is considered a second order method. Since the Fisher Information Matrix (FIM) is positive semi-definite, the Hessian term is also positive semi-definite, ensuring a convex optimization.

Meanwhile, Newton's method is a second-order optimization method that is effective only when the loss function is strongly convex with a Lipschitz continuous Hessian. The update equation for Newton's method Bonnans et al. (2006) is as follows.

$$\boldsymbol{\theta}_{t+1} = \boldsymbol{\theta}_t - \eta \boldsymbol{H}_{\boldsymbol{\theta}}(\mathcal{L}(\boldsymbol{\theta}))^{-1} \nabla \mathcal{L}(\boldsymbol{\theta}) \tag{50}$$

Although the two methods were derived through vastly different processes, they both involve the inverse of the Hessian matrix, as shown in Eq. (49) and Eq. (50). However, there is a significant difference between the two methods. Natural gradient descent requires the loss function to be the log-likelihood, while Newton's method has no such restriction on the choice of loss function. Natural gradient descent guarantees convergence due to the positive semi-definiteness of FIM (the expected value of the Hessian), whereas Newton's method does not.

Consequently, various complex techniques are employed when using Newton's method, such as the Gauss-Newton algorithm, conjugate gradient method, and trust region method Bonnans et al. (2006). These techniques ensure that each step update is confined within a trust region, mitigating the risk of divergence.

This is why second-order optimization methods based on FIM have demonstrated faster convergence compared to Newton's method in practice (Schraudolph, 2002; Martens et al., 2010; Vinyals & Povey, 2012). Recent second-order optimization methods often utilize FIM instead of the Hessian of the loss function Gupta et al. (2018); Anil et al. (2019). However, these methods require the loss function to be the log-likelihood and are subject to certain constraints discussed in Section 3.2 and Section 3.3.

### B.2 CONDITIONAL PROBABILITY DISTRIBUTION FOR EMPIRICAL FISHER INFORMATION

In supervised learning, the loss function typically becomes a joint distribution $-\log P(\mathrm{x}, \mathrm{y}|\boldsymbol{\theta})$. Since the expected value over $P(\mathrm{x}, \boldsymbol{\theta})$ is generally intractable, it is replaced with the empirical distribution $\hat{p}_{data}(\mathrm{x})$ in Eq. (52). As noted in Eq. (16), this is referred to as the empirical Fisher. The training set also provides the ground truth label y. Therefore, the loss function becomes the cross-entropy calculated using the conditional log-likelihood, as shown in Eq. (53).

$$\nabla_{\boldsymbol{\theta}} J(\boldsymbol{\theta}) = -\mathbb{E}_{\mathrm{x}, \mathrm{y} \sim p_{data}}[\nabla_{\boldsymbol{\theta}} \log P(\mathrm{y}|\mathrm{x}, \boldsymbol{\theta}) + \nabla_{\boldsymbol{\theta}} \log P(\mathrm{x}|\boldsymbol{\theta})] \tag{51}$$

$$\approx -\mathbb{E}_{\mathrm{x}, \mathrm{y} \sim \hat{p}_{data}}[\nabla_{\boldsymbol{\theta}} \log P(\mathrm{y}|\mathrm{x}, \boldsymbol{\theta}) + \cancel{\nabla_{\boldsymbol{\theta}} \log \hat{p}_{data}(\mathrm{x})}] \tag{52}$$

$$= -\mathbb{E}_{\mathrm{x}, \mathrm{y} \sim \hat{p}_{data}}[\nabla_{\boldsymbol{\theta}} \log P(\mathrm{y}|\mathrm{x}, \boldsymbol{\theta})] := \boldsymbol{g} \tag{53}$$

The FIM, similar to the cost function, also requires the calculation of an expected value over the joint distribution. By approximating the expectation with the empirical distribution $\hat{p}_{data}(\mathrm{x}, \mathrm{y})$ and moving out the square as shown in Eq. (21), we obtain the following equation:

$$\hat{\boldsymbol{f}}(\boldsymbol{\theta}) = \mathbb{E}_{\mathrm{x}, \mathrm{y} \sim P(\mathrm{y}|\mathrm{x}, \boldsymbol{\theta})P(\mathrm{x}|\boldsymbol{\theta})}[(\nabla_{\boldsymbol{\theta}} \log P(\mathrm{y}|\mathrm{x}, \boldsymbol{\theta}) + \nabla_{\boldsymbol{\theta}} \log P(\mathrm{x}|\boldsymbol{\theta}))^2] \tag{54}$$

$$\approx \mathbb{E}_{\mathrm{x}, \mathrm{y} \sim P(\mathrm{y}|\mathrm{x}, \boldsymbol{\theta})\hat{p}_{data}(\mathrm{x})}[(\nabla_{\boldsymbol{\theta}} \log P(\mathrm{y}|\mathrm{x}, \boldsymbol{\theta}) + \cancel{\nabla_{\boldsymbol{\theta}} \log \hat{p}_{data}(\mathrm{x})})^2] \tag{55}$$

$$\approx \mathbb{E}_{\mathrm{x}, \mathrm{y} \sim P(\mathrm{y}|\mathrm{x}, \boldsymbol{\theta})\hat{p}_{data}(\mathrm{x})}[\nabla_{\boldsymbol{\theta}} \log P(\mathrm{y}|\mathrm{x}, \boldsymbol{\theta})^2] \tag{56}$$

As shown in equation Eq. (22), generative models are able to reuse the gradient $\boldsymbol{g}$ for calculating FIM. However, this reuse poses a challenge when dealing with conditional distributions. This is

because Equation Eq. (53) calculates the expected value over the label y, while Equation Eq. (56) calculates the expected value over the conditional distribution of the model. To address this, we introduce an additional approximation by calculating the expected value over the label y in Eq. (57). This is commonly referred to as the **empirical FIM** in the ML community (Kunstner et al., 2019), while the statistics community refers to the approximation in Eq. (55) as the empirical FIM, as explained in Eq. (16). To avoid confusion, we will refer to this as the **conditional empirical FIM**.

$$\hat{f}(\boldsymbol{\theta}) \approx \mathbb{E}_{\mathrm{x,y} \sim \hat{p}_{data}}[\nabla_{\boldsymbol{\theta}} \log P(\mathrm{y}|\mathrm{x}, \boldsymbol{\theta})^2] \tag{57}$$

$$\approx \mathbb{E}_{\mathrm{x,y} \sim \hat{p}_{data}}[\nabla_{\boldsymbol{\theta}} \log P(\mathrm{y}|\mathrm{x}, \boldsymbol{\theta})]^2 = \boldsymbol{g}^2 \tag{58}$$

As seen in Eq. (21), to reuse $\boldsymbol{g}$ in Eq. (53), a non-principled approximation is made by taking the square outside the expectation in Eq. (58). Similar to the generative model in Eq. (22), we can obtain the natural gradient with respect to a minibatch $\mathcal{B}$ as follows:

$$\tilde{\nabla} J(\boldsymbol{\theta}) = \hat{f}^{-1} \nabla J(\boldsymbol{\theta}) \approx -\mathbb{E}_{\mathrm{x,y} \in \mathcal{B}}[\nabla_{\boldsymbol{\theta}} \log P(\mathrm{y}|\mathrm{x}, \boldsymbol{\theta})]/\mathbb{E}_{EMA}[\mathbb{E}_{\mathrm{x,y} \in \mathcal{B}}[\nabla_{\boldsymbol{\theta}} \log P(\mathrm{y}|\mathrm{x}, \boldsymbol{\theta})]^2] \tag{59}$$

$$= -\boldsymbol{g}/\mathbb{E}_{EMA}[\boldsymbol{g}^2] \tag{60}$$

Despite the numerous non-trivial approximations made to FIM, it is remarkable that Adam has achieved such great success. Eliminating these non-trivial approximations could be a promising direction for future research.

### B.3 CLIPPING AND EPSILON

Although not mentioned in the original Adam paper, it was later discovered that gradient clipping is essential for Adam Zhang et al. (2020); Gilmer et al. (2021) and is now used as a standard practice. Adafactor Shazeer & Stern (2018) incorporates clipping Pascanu et al. (2013) in the algorithm, because the diagonal empirical FIM can have zero components. Therefore, **clipping is applied to the invariant natural gradient**, and then the clipped gradient is used to calculate the momentum, as demonstrated in Algorithm 1. The Adafactor implementation Shazeer (2018) already incorporates this approach, although it is not explicitly mentioned in the original paper Shazeer & Stern (2018), as shown in Algorithm 4. Since clipping is already incorporated into the optimizer, global clipping Pascanu et al. (2013) is not necessary.[18] In our experiments, we did not observe any benefit from applying global clipping.

To prevent division by zero before clipping, $\epsilon$ is added. The standard value for $\epsilon$ in Adam is 1e-8. Adafactor Shazeer & Stern (2018) uses 1e-30, but since it is added inside the square root, this translates to 1e-15 on the Adam scale. Wortsman et al. (2023) recommends using a smaller epsilon value of 1e-15 for large-scale models like LLMs due to the empirical observation that the root mean square (RMS) value of gradients tends to decrease as models get larger and training progresses.

To prevent division by zero before clipping, $\epsilon$ is added. The standard value for $\epsilon$ in Adam is 1e-8. However, Wortsman et al. (2023) recommends using a smaller epsilon value of 1e-15 for large-scale models like LLMs. This is because the root mean square (RMS) value of gradients tends to decrease as models get larger and training progresses. Adafactor Shazeer & Stern (2018) uses 1e-30, but since it is added inside the square root, this translates to 1e-15 on the Adam scale. In our experiments, we confirm that an $\epsilon$ value of around 1e-15 yields good results across all domains, as shown in Appendix C.1.2. Therefore, we use 1e-15 as the default $\epsilon$ for FAdam, which provides both good performance and robustness for large models.

EAdam Yuan & Gao (2020) and Adafactor Shazeer & Stern (2018) modified the algorithm to accumulate $\epsilon$ within EMA, but we did not observe any benefits from this approach in our experiments.

### B.4 FADAFACTOR

Adafactor Shazeer & Stern (2018) is modified as shown in Algorithm 2, and we refer to this variant as Fisher Adafactor (FAdafactor).

---

[18]This feature is often included in ML frameworks as a legacy practice without a clear justification.

**Algorithm 2** Fisher Adafactor (FAdafactor)

1: **given** $\beta_1 = 0.9$, $\beta_2 = 0.999$, $\epsilon = 10^{-15}$, $c = 1$, $\lambda = 0.001$, $\rho = 0.5$, $\eta_t$
2: **initialize** $\boldsymbol{\theta}_0$, $t \leftarrow 0$, $\boldsymbol{m}_0 \leftarrow 0_{n \times m}$, $\boldsymbol{R}_0 \leftarrow 1_n$, $\boldsymbol{C}_0 \leftarrow 1_m^\top$ ▷ FIM init to 1 as per Section 3.4.4
3: **repeat**
4:     $t \leftarrow t + 1$
5:     $\boldsymbol{g}_t \leftarrow \nabla_{\boldsymbol{\theta}} \log P_t(\boldsymbol{\theta}_{t-1})$           ▷ Stochastic gradient $\boldsymbol{g}_t \in \mathbb{R}^{n \times m}$ as per Eq. (12)
6:     $\hat{\beta}_2 \leftarrow \beta_2(1 - \beta_2^{t-1})/(1 - \beta_2^t)$         ▷ Bias correction as per Section 3.4.4
7:     $\boldsymbol{R}_t \leftarrow \hat{\beta}_2 \boldsymbol{R}_{t-1} + (1 - \hat{\beta}_2)\boldsymbol{g}_t^2 1_m$         ▷ EMA column vector $\boldsymbol{R}_t \in \mathbb{R}^n$
8:     $\boldsymbol{C}_t \leftarrow \hat{\beta}_2 \boldsymbol{C}_{t-1} + (1 - \hat{\beta}_2)1_n^\top \boldsymbol{g}_t^2$         ▷ EMA row vector $\boldsymbol{C}_t \in \mathbb{R}^m$
9:     $\boldsymbol{f}_t \leftarrow \boldsymbol{R}_t \boldsymbol{C}_t / 1_n^\top \boldsymbol{R}_t$         ▷ Diagonal empirical FIM as per Section 3.4.1
10:    $\bar{\boldsymbol{g}}_t \leftarrow \boldsymbol{g}_t / (\boldsymbol{f}_t^\rho + \epsilon)$         ▷ Invariant natural gradient as per Eq. (27)
11:    $\bar{\boldsymbol{g}}_t \leftarrow \bar{\boldsymbol{g}}_t / \max(1, \text{RMS}(\bar{\boldsymbol{g}}_t)/c)$         ▷ Clip the gradient as per Appendix B.3
12:    $\boldsymbol{m}_t \leftarrow \beta_1 \boldsymbol{m}_{t-1} + (1 - \beta_1)\bar{\boldsymbol{g}}_t$         ▷ EMA momentum as per Section 3.4.2
13:    $\bar{\boldsymbol{g}}_w \leftarrow \boldsymbol{\theta}_{t-1} / (\boldsymbol{f}_t^\rho + \epsilon)$         ▷ Weight decay as per Eq. (28)
14:    $\bar{\boldsymbol{g}}_w \leftarrow \bar{\boldsymbol{g}}_w / \max(1, \text{RMS}(\bar{\boldsymbol{g}}_w)/c)$         ▷ Clip weight decay as per Appendix B.3
15:    $\boldsymbol{\theta}_t \leftarrow \boldsymbol{\theta}_{t-1} - \eta_t(\boldsymbol{m}_t + \lambda \bar{\boldsymbol{g}}_w)$         ▷ Update $\boldsymbol{\theta}$ as per Eq. (13)
16: **until** *stopping criterion is met*
17: **return** optimized parameters $\boldsymbol{\theta}_t$

## B.5 ADAM AND ADAFACTOR

To facilitate comparison between the original algorithms, we present Adam (Algorithm 3) and Adafactor (Algorithm 4), alongside FAdam (Algorithm 1) and FAdafactor (Algorithm 2). Since the full Adafactor algorithm is not explicitly detailed in the original paper Shazeer & Stern (2018), we refer the original implementation Shazeer (2018).

Remarkably, Adafactor Shazeer & Stern (2018) had already empirically discovered and implemented several of the results we derived from our mathematical framework: applying natural gradient to momentum (Section 3.4.2), omitting bias correction for momentum (Section 3.4.2), and clipping gradients (Appendix B.3). This is akin to the invention of the steam engine before the establishment of thermodynamics, and these empirical findings significantly bolstered our confidence in developing our theory. However, it is peculiar that the momentum calculation incorporates the learning rate, while weight decay remains independent of the learning rate. Accumulating $\epsilon$ through EMA is also an unconventional approach.

**Algorithm 3** Adam Kingma & Ba (2014)

1: **given** $\beta_1 = 0.9$, $\beta_2 = 0.999$, $\epsilon = 10^{-8}$, $c = 1$, $\lambda = 0.001$, $\eta_t$
2: **initialize** $\boldsymbol{\theta}_0$, $t \leftarrow 0$, $\boldsymbol{m}_0 \leftarrow 0_N$, $\boldsymbol{f}_0 \leftarrow 0_N$
3: **repeat**
4:     $t \leftarrow t + 1$
5:     $\boldsymbol{g}_t \leftarrow \nabla_{\boldsymbol{\theta}} \log P_t(\boldsymbol{\theta}_{t-1})$
6:     $\boldsymbol{f}_t \leftarrow (\beta_2 \boldsymbol{f}_{t-1} + (1 - \beta_2)\boldsymbol{g}_t^2)/(1 - \beta_2^t)$
7:     $\boldsymbol{m}_t \leftarrow (\beta_1 \boldsymbol{m}_{t-1} + (1 - \beta_1)\boldsymbol{g}_t)/(1 - \beta_1^t)$
8:     $\bar{\boldsymbol{g}}_t \leftarrow \boldsymbol{m}_t / (\sqrt{\boldsymbol{f}_t} + \epsilon)$
9:     $\bar{\boldsymbol{g}}_t \leftarrow \bar{\boldsymbol{g}}_t / \max(1, \text{RMS}(\bar{\boldsymbol{g}}_t)/c)$
10:    $\boldsymbol{\theta}_t \leftarrow \boldsymbol{\theta}_{t-1} - \eta_t(\bar{\boldsymbol{g}}_t + \lambda \boldsymbol{\theta}_{t-1})$
11: **until** *stopping criterion is met*
12: **return** optimized parameters $\boldsymbol{\theta}_t$

## B.6 CONVERGENCE PROOF

While the original Adam paper Kingma & Ba (2014) presented a convergence proof, Reddi et al. (2019) later demonstrated that this proof was flawed. Subsequently, Défossez et al. (2020) provided a corrected convergence proof for Adam. Our analysis follows the approach of Défossez et al. (2020).

---

**Algorithm 4** Adafactor Shazeer & Stern (2018); Shazeer (2018)

---

1: **given** $\beta_1 = 0.9$, $\beta_2 = 0.999$, $\epsilon = 10^{-30}$, $c = 1$, $\lambda = 0.001$, $\eta_t$
2: **initialize** $\boldsymbol{\theta}_0$, $t \leftarrow 0$, $\boldsymbol{m}_0 \leftarrow 0_{n \times m}$, $\boldsymbol{R}_0 \leftarrow 0_n$, $\boldsymbol{C}_0 \leftarrow 0_m^\top$
3: **repeat**
4:     $t \leftarrow t + 1$
5:     $\boldsymbol{g}_t \leftarrow \nabla_{\boldsymbol{\theta}} \log P_t(\boldsymbol{\theta}_{t-1})$
6:     $\hat{\beta}_2 \leftarrow \beta_2 (1 - \beta_2^{t-1})/(1 - \beta_2^t)$
7:     $\boldsymbol{R}_t \leftarrow \hat{\beta}_2 \boldsymbol{R}_{t-1} + (1 - \hat{\beta}_2)(\boldsymbol{g}_t^2 + \epsilon 1_n 1_m^\top) 1_m$
8:     $\boldsymbol{C}_t \leftarrow \hat{\beta}_2 \boldsymbol{C}_{t-1} + (1 - \hat{\beta}_2) 1_n^\top (\boldsymbol{g}_t^2 + \epsilon 1_n 1_m^\top)$
9:     $\boldsymbol{f}_t \leftarrow \boldsymbol{R}_t \boldsymbol{C}_t / 1_n^\top \boldsymbol{R}_t$
10:     $\bar{\boldsymbol{g}}_t \leftarrow \boldsymbol{g}_t / \sqrt{\boldsymbol{f}_t}$
11:     $\bar{\boldsymbol{g}}_t \leftarrow \bar{\boldsymbol{g}}_t / \max(1, \text{RMS}(\bar{\boldsymbol{g}}_t)/c)$
12:     $\boldsymbol{m}_t \leftarrow \beta_1 \boldsymbol{m}_{t-1} + (1 - \beta_1) \eta_t \bar{\boldsymbol{g}}_t$
13:     $\boldsymbol{\theta}_t \leftarrow \boldsymbol{\theta}_{t-1} - (\boldsymbol{m}_t + \lambda \boldsymbol{\theta}_{t-1})$
14: **until** *stopping criterion is met*
15: **return** optimized parameters $\boldsymbol{\theta}_t$

---

To the best of our knowledge, there is no existing convergence proof for Adam with clipping, despite its widespread use in practice. This is likely due to the non-trivial nature of incorporating clipping into the analysis. Existing convergence proofs only cover SGD with clipping Mukherjee & Tucat (2024). Therefore, we also present a convergence proof that excludes the effects of clipping.

The convergence proof for FAdam is simpler than that of Adam. Adam's convergence proof needs to consider both the 1st and 2nd momentums simultaneously, as they both influence the gradient update. In contrast, FAdam first computes the natural gradient using FIM and then applies momentum to the natural gradient. This process of applying momentum to the natural gradient is analogous to Polyak momentum applied to SGD. It is well-known that Polyak momentum tightens the convergence bound Bottou et al. (2018). Since FAdam's momentum is analogous to Polyak momentum, FAdam's momentum also tightens the convergence bound. Therefore, the convergence bound for the natural gradient without momentum is looser than the convergence bound for FAdam. For the sake of simplicity, we focus on deriving the convergence bound for the natural gradient without momentum. A tighter theoretical convergence bound, incorporating the effects of momentum, is left for future work.

FAdam without the effects of momentum is equivalent to the Adam algorithm with $\beta_1 = 0$. The convergence of Adam under this condition is shown in Eq. (15) of Défossez et al. (2020) as follows:

$$\mathbb{E}\left[\|\nabla F(x_\tau)\|^2\right] \leq \frac{F(x_0) - F_*}{\alpha_1 \sqrt{N}} + \frac{1}{\sqrt{N}}(4dR^2 + \alpha_1 dRL)\left(\ln\left(1 + \frac{RN}{\epsilon}\right) + \frac{N}{N-1}\right) \quad (61)$$

$$\sim O(d \ln(N)/\sqrt{N}) \quad (62)$$

In Eq. (61), $x \in \mathbb{R}^d$ represents the model parameters $\boldsymbol{\theta}$, $\alpha_1$ is the learning rate, $N$ represents the number of iterations, and $\tau_N$ denotes a random index within the set $\{0, \ldots, N-1\}$. To derive the equation, the following assumptions are made:

The function $F$ is bounded below by $F_*$, that is:

$$F(x) \geq F_* \text{ for all } x \in \mathbb{R}^d.$$

The $l_\infty$ norm of the stochastic gradients is uniformly almost surely bounded, meaning there exists $R \geq \sqrt{\epsilon}$ such that:

$$\|\nabla F(x)\|_\infty \leq R - \sqrt{\epsilon} \text{ for all } x \in \mathbb{R}^d.$$

The objective function is smooth, specifically, its gradient is L-Lipschitz continuous with respect to the $l_2$-norm:

$$\|\nabla F(x) - \nabla F(y)\|_2 \leq L\|x - y\|_2 \text{ for all } x, y \in \mathbb{R}^d.$$

Therefore, from Eq. (62), as the number of iterations $N$ increases, the expected value of the squared norm of the gradient diminishes, indicating that $F$ converges to the optimal value $F_*$.

# C  EXPERIMENT

## C.1  ABLATION STUDY

### C.1.1  EXPONENT OF FIM

In Section 3.4.1, we provided a theoretical explanation for the use of the square root in FIM, as shown in Eq. (27). To explore alternative exponent values, we conducted experiments, the results of which are presented in Fig. 1d and Table 2. Exponents above 0.5 exhibit relative stability, while values below 0.3 demonstrate a sharp decline in performance. The standard value of 0.5 for the FIM exponent in Adam Kingma & Ba (2014) appears to be a well-justified choice.

| Model | Metric | Steps | $\rho$=0.3 | $\rho$=0.4 | $\rho$=0.5 | $\rho$=0.6 | $\rho$=0.7 | $\rho$=0.8 | $\rho$=0.9 | $\rho$=1.0 |
|---|---|---|---|---|---|---|---|---|---|---|
| LLM 1B | loss | 1.5k | 5.4 | 3.9 | **3.8** | 3.9 | 4.2 | 4.3 | 4.3 | 4.3 |
| ASR 100M | WER | 8k | 43.9 | 11.2 | **6.04** | 6.08 | **6.04** | 6.08 | 6.23 | 6.33 |

Table 2: Effects of Varying FIM Exponent on LLM and ASR Model Performance

### C.1.2  EPSILON

Our experiments examined the effect of different epsilon values on the performance of LLM (1B), ASR (600M), and VQ-VAE (100M) models. We observed that the optimal epsilon value varied across models of different sizes and domains. However, optimal values tended to be near 1e-15. In some domains, extremely small epsilon values (1e-20 or 1e-30) caused training to diverge.

| Model | Metric | Steps | 1e-8 | 1e-12 | 1e-15 | 1e-20 | 1e-30 |
|---|---|---|---|---|---|---|---|
| LLM 1B | loss | 80k | 2.47 | **2.42** | 2.46 | NaN | NaN |
| ASR 600M | WER | 8k | 2.08 | **2.05** | 2.08 | **2.05** | 2.13 |
| VQ-VAE 100M | FID | 50k | 20.9 | 19.9 | **19.6** | 19.7 | NaN |

Table 3: Impact of Epsilon on Model Performance (Lower is Better)

### C.1.3  WEIGHT DECAY

As shown in Eq. (28), the most significant algorithmic change in FAdam compared to Adam is the weight decay mechanism. Therefore, we conducted experiments for weight decay.

In the ASR experiments described in Section 4.2, the baseline model utilized Adam, and we found that the weight decay parameter ($\lambda$) used in Adam could generally be reused for FAdam. Our enhanced weight decay mechanism played a significant role in improving FAdam's performance, as shown in Table 4.

| LibriSpeech WERs | dev | dev-other | test | test-other | avg |
|---|---|---|---|---|---|
| Adam $\lambda$=0 | 1.31 | 2.55 | 1.34 | 2.57 | 1.94 |
| Adam $\lambda$=0.001 | 1.30 | 2.54 | 1.33 | 2.59 | 1.93 |
| FAdam $\lambda$=0 | 1.30 | 2.43 | 1.34 | 2.63 | 1.92 |
| **FAdam $\lambda$=0.001** | 1.29 | 2.49 | 1.34 | 2.49 | **1.89** |

Table 4: LibriSpeech WERs of 600M ASRs with Varying Weight Decay Parameter ($\lambda$)

In the LLM experiments described in Section 4.1, the baseline model utilized Adafactor, which decouples weight decay from the learning rate, as shown in Algorithm 4. Interestingly, despite the fact that Adafactor and FAdafactor have quite different weight decay mechanisms, the optimal weight decay value of 1e-3 for Adafactor also is optimal for FAdafactor in Table 5.

| $\lambda$ | 1e-2 | 1e-3 | 1e-4 | 1e-5 |
|---|---|---|---|---|
| LLM 1B | 3.007 | **2.875** | 2.998 | 3.004 |

Table 5: Eval loss of 1B LLMs with Varying Weight Decay Parameter ($\lambda$)

## C.2 EXPERIMENTS COMPUTE RESOURCES

FAdam and Adam exhibited nearly identical computational and memory complexity. During TPU training, both optimizers achieved very similar steps/sec.

The LLM 1B model was trained on 16 TPUv5 Jouppi et al. (2023) devices (80GB HBM) for one day. Each training example consisted of 2k tokens with a global batch size of 256. The ASR 600M model was fine-tuned on 32 TPUv3 devices (8GB HBM) for half a day. Each utterance had an average duration of 12 seconds with a global batch size of 512. The VQ-VAE 100M model was trained on 16 TPUv4 devices (16GB HBM) for one day. Training was performed using images with a resolution of 256x256 and a global batch size of 256.

