# OpenReview forum: "FAdam: Adam is a natural gradient optimizer using diagonal empirical Fisher information"
_ICLR.cc/2025/Conference — ICLR 2025 Conference Withdrawn Submission_

### Official Review · Reviewer_FbYu · 2024-10-21

**Soundness:** 2
**Presentation:** 1
**Contribution:** 1
**Rating:** 3
**Confidence:** 3

**Summary:**

In this paper, the authors provide an explanation of the second-order moment $v_t$ of Adam from the perspective of the diagonal Fisher information, and propose a new optimizer, FAdam, by utilizing this perspective.

**Strengths:**

This paper provides a comprehensive discussion of the previous works, and the empirical results seem supportive.

**Weaknesses:**

At least from my perspective, I can not grasp the main contribution of this paper. I feel it is more like a technical report or a review instead of a paper. I will list my concerns as follows:

1. The main conclusion of this paper is somewhat unclear, and the writing is difficult to follow. From my understanding, the authors aim to claim that Adam is a variant of natural gradient descent and introduce a new optimizer, Fadam, as I outlined in the summary. However, they spend over half of the paper discussing basic statistical properties and formulas related to Fisher information, along with extensive reviews of previous works, without presenting their own results or conclusions. In contrast, the descriptions of the algorithms and the theoretical convergence results are glossed over. While I acknowledge that some discussion of prior works is necessary, I believe it should be integrated with the proposed methods and conclusions of this paper. In summary, the lengthy review of existing literature and preliminary knowledge renders the current manuscript confusing and unappealing.


2. The technical contribution of this paper is relatively insufficient. As a work proposing a new optimizer, the authors fail to provide a rigorous theoretical guarantee of convergence. The current version's convergence analysis disregards the effects of momentum, and even this incomplete result is derived directly from another paper. Furthermore, the statement that _“Since FAdam’s momentum is analogous to Polyak momentum, FAdam’s momentum also tightens the convergence bound. Therefore, the convergence bound for the natural gradient without momentum is looser than the convergence bound for FAdam,”_ is presented without adequate justification and is not convincing. It is unreasonable to assert that the convergence bound of one optimizer is looser than that of another without rigorous derivation.

**Questions:**

I suggest the authors rethink their major contribution of this paper. As a paper to propose a new optimizer, it might be better to first introduce the new algorithm and present the pros of this algorithm (they could be empirical results of theoretical guarantees). Although I understand some basic illustrations about the preliminary knowledge or motivation of some terms are necessary, at least for this paper, I believe the discussion in the current manuscript should be refined. For example,  I do not get any interesting insights from the discussion about the connection between log-likelihood of Gaussian distribution and $\ell_2$ loss, as it is a basic knowledge of statistics, and seems not deeply correlated with the Fadam.

---

### Official Review · Reviewer_PpE9 · 2024-11-03

**Soundness:** 2
**Presentation:** 3
**Contribution:** 2
**Rating:** 3
**Confidence:** 2

**Summary:**

The authors proposed a connection between the Adam optimizer and natural gradient optimization, treating the moving average of squared gradients in Adam as an estimate of the diagonal elements of the Fisher information matrix. They hypothesised that Adam's advantage over other methods might be due to its use of natural gradients; the advantage is particularly noticeable in tasks with discrete distributions, since they allow for a tighter approximation of the Fisher matrix.

The authors also offered a justification for the necessity of normalization by the square root of squared gradients to ensure basis invariance when averaging gradients in Adam. Additionally, they analyzed how momentum, weight decay, and clipping should function in the context of natural gradients and proposed new variants of Adam and Adafactor — FAdam and FAdafactor. The proposed FAdam method demonstrates superior performance for models like LLMs and VQ-VAEs, as well as in ASR tasks.

**Strengths:**

1. The authors have identified an important and interesting connection between the success of the Adam optimization method and optimization by the natural gradient method.
2. The authors proposed an explanation for why Adam's advantages primarily emerge in problems with discrete distributions.
3. The authors established principles for using momentum, weight decay, and clipping in optimization with invariant gradients. Based on this analysis, they proposed a new method — FAdam, which demonstrates improved performance compared to traditional Adam.

**Weaknesses:**

1. The authors did not provide an analysis to assess the accuracy of the approximations and simplifications used in this method.

a) Why is the transition from sampling from $p(x|\theta)$ to sampling from $p_{data}$ valid? In the case of an undertrained model, the distribution $p(x|\theta)$ can differ significantly from the marginalized $p_{data}$ distribution.

b) How accurate is the transition from Eq. (20) to Eq. (21) and why it is not critical to the method's effectiveness?

c) How accurate is the FIM approximation throughout the hundreds of optimization steps for EMA?

If the authors could provide ablation experiments comparing Adam, FAdam, a true natural gradient method, and other methods incorporating intermediate transitions on simple tasks (e.g., CIFAR-10), it would significantly increase confidence in the results.

2. In the theoretical justification for preferring discrete distributions: due to uniform sampling from $p_{data}$, discrete distributions can also provide a poor approximation of the FIM. This is because the concentration of the distribution may lie in false logits, which is common in yet-not-fully-trained networks. The score might not be large enough to yield a good approximation.

3. While Amari et al. (2019) prove that unit-wise block diagonal FIM has off-diagonal blocks smaller by $\frac{1}{\sqrt{n}}$, the authors' interpretation appears to extend beyond the original result. Their derived claim about individual diagonal weights dominating off-diagonal weights by $\frac{1}{\sqrt{n}}$ (lines 186-188) may need additional justification, as it's not directly supported by Amari's work.

4. The absence of confidence intervals in the experimental results prevents from being fully certain of FAdam superiority, due to marginal score improvements. Additionally, providing further experiments on a broader range of domains would strengthen the evidence of the proposed method's improved performance.

5. Minor typos: in Eq. (25), it should be the square of the norm; in Eq. (19), "approx." should replace the equality sign.

To summarize, I believe this paper relies too heavily on unjustified approximations and is not yet ready for the conference. However, if the authors provide additional experimental and theoretical validation, I might increase my score.

**Questions:**

Please see the weaknesses for questions and improvements.

---

### Official Review · Reviewer_yRgP · 2024-11-04

**Soundness:** 1
**Presentation:** 2
**Contribution:** 2
**Rating:** 3
**Confidence:** 4

**Summary:**

This work takes a statistical viewpoint of the Adam optiumization algorithm and attempts to both explain it’s performance and add improvlemtents through the lense of the natural graident algorithm. The authors argue that Adam is effectivly preconditioning  with the diagonal of the Fisher Information Matrix,  which leads to  the  algorithm’s superior performance.

**Strengths:**

The approach of  analyzing Adam from a statistical  viewpoint is interesting, and while not being new (this interpretation was mentioned in  the origional  Adam paper) it could  deserve a second  look. The authors additionally show some empiracle improvments in a few settings.

**Weaknesses:**

While section 2.1 is likely relevant for  doing a detailed  analysis of Adam in the proposed framework, as far  as I can tell that analysis does not actually take place in the paper or appendix. Given this this  section  feels quite out of place to me? I’m unsure of  it’s value for the main message of the paper. Most readers in the optimization or  statistics community are familiar with Fisher Information Matrix based methods and their connections to second order newton style algorithms,  so I’m unsure of the value of introducing them using ideas form differential geometry on manifolds.

Throughout this paper is repeatedly claimed that Adam is preconditioning with the diagonal  of the Fisher Information Matrix, the approximation used in Adam is not the same thing  in general. Adam has been connected  to second order like or natural gradient like algorithms, but it is known in general that the gradient squared is not an approximation to  the diagonal of the Fisher Information Matrix. While it is in expectation, in the finite data regime there  are clear counter examples to this, such as in [1]  which the author cites.

On line  250 the authors claim that Adam excels  in classification tasks such as next  token prediction, but this seems somewhat  contradictory to the previous line where it claimed that CNNs are often better when trained with SGD. I agree that in many vision tasks SGD matches Adam in performance, but what is left out of  the text is that that is true in most classification problems  in vision, which is again a discrete output space. I’m additionally not  sure   of the strength of the claim that Adam is less strong in the generative  setting. I’m aware of works  such as  [2] that claim the oposite (which the author cites) attempting to answer the question “What factors explain that Adam produces better quality solutions than SGDA when training GANs”, and propose modifications to SGD to help it compete with Adam.

Overall several assertions are made that Adam fails on continuous regression targets, but I feel like there  is not sufficient citation or experimentation to back that up. Adam excelling in discrete output spaces (which again is not always true, training  ResNets with SGD is still very common) is not the same thing as Adam failing on continuous tasks, and needs to be justified if it is being claimed. Examples counter to this idea exist in the literature, such as the quadratic function minimized in figure 6 of [3] where Adam handily outperforms gradient descent.

The notation of some of the equations is a but unclear, for example in equation 15 while I understand the division is being coordinate  wise, this should be explicit in the notation,  otherwise a less familiar reader may thing we’re trying to divide a vector by another vector which is ill defined.

Cosmetically, the citation style is very   non-standard and makes reading difficult, I would suggest the authors use a  more standard method of in text citation.

Minor but Adam Algorithm written in B.5 is in fact AdamW and has clipping added which was  not included in the origional algorithm.

The central weakness of this paper in my opinion is it misunderstands how approximate the approximations in Adam are. The idea of the Adam update being connected to the diagonal of the Fisher information matrix is not new, it was mentioned in Kingma and Ba (2014). The optimization community has tried very hard to understand why Adam works (another weakness of this paper is there is no related work regarding the vast amount of research into understanding Adam) and this approach has not appeared to yield progress. The authors acknowledge the significance of these approximations in appendix B.3, but given the amount of work showing that these approximations are often very poor I don’t think the community can comfortably understand Adam as a natural gradient algorithm.


[1]
Frederik Kunstner, Lukas Balles, Philipp Hennig

Limitations of the Empirical Fisher Approximation for Natural Gradient Descent

https://arxiv.org/abs/1905.12558

[2]
Samy Jelassi, Arthur Mensch, Gauthier Gidel, Yuanzhi Li\

Adam is no better than normalized SGD: Dissecting how adaptivity improves GAN performance

https://openreview.net/pdf?id=D9SuLzhgK9

[3]
Frederik Kunstner, Robin Yadav, Alan Milligan, Mark Schmidt, Alberto Bietti

Heavy-tailed class imbalance and why adam outperforms gradient descent on language models

https://arxiv.org/pdf/2402.19449

**Questions:**

- Can the authors give a citation regarding using an EMA for fisher info on line 228? I’m not aware that’s been used prior to Adam.
- How does the proposed framework justify clipping? In B.3 clipping  and epsilon is mentioned  through related work, but this step that has been added to the algorithm does not apppear to be justified by the theoretical framework.
- What norms are being used in equations (25)-(27)? I’m assuming the first norm is the one induced by the Fisher Information Matrix, but then what is the other one? Euclidian?
- Has the author tried to quantify how accurate the approximations (F)Adam is using are in a simple setting? This can help figure out if those approximations are in fact reasonable, which needs to be the case in order to claim it’s really natural gradient in disguise.

---

### Official Review · Reviewer_ZZV1 · 2024-11-06

**Soundness:** 2
**Presentation:** 2
**Contribution:** 2
**Rating:** 3
**Confidence:** 4

**Summary:**

The paper reiterates and expands the motivation of Adam as approximate natural gradient descent. It derives multiple modifications to the Adam algorithm based on that interpretation. The resulting method (FAdam) is evaluated experimentally.

**Strengths:**

- The argument relating decoupled weight decay to the information-geometric interpretation is interesting. It clarifies that the gradients used to compute $v$ (the diagonal empirical FIM) must be gradients of the log likelihood of a probabilistic model to match the definition of the FIM and therefore must not contain regularizers or auxiliary losses.
- Averaging the preconditioned gradients (versus preconditioning the averaged gradient) is an interesting variant.

**Weaknesses:**

- The paper presents as an original finding that it "establishes a mathematical foundation for the Adam optimizer" in terms of NGD with the empirical Fisher information matrix. This is misleading. This motivation has been given in the original Adam paper and has since been discussed and critiqued in various papers, including but not limited to Kunstner et al. (2019). This should be made transparent in the discussion of related work.
- The paper states that "for using natural gradient optimizers [...] the loss function must be in the form of the log-likelihood". This is not a factual statement and should be adjusted. Preconditioning with the Fisher information matrix adapts to the geometry induced by a certain probabilistic model. The negative log likelihood under said model may be a "natural" objective function to optimize, but NGD can meaningfully be applied to any other objective. In fact, in Section 3.4.3, the authors advocate for preconditioning an additional loss term with the FIM.
- The argument in Section 3.4.1 regarding the use of the square-root on the preconditioner is not stringent. If I am understanding correctly, the argument is that $\Vert \nabla J/\sqrt{f} \Vert^2_2 \approx \Vert F^{-1} \nabla J\Vert_F^2$, i.e., preconditioning with the square-root makes the Euclidean norm of the resulting update equal the  "Fisher norm" of the natural gradient. However, there is no discernible argument why it would be desirable to match these two quantities and, if so, why one would want to achieve this by changing the preconditioner rather than, say, scale the update with a scalar factor? (Minor: The notation should also be improved in Eq. (25) - (27), since $\Vert\cdot\Vert$ is used to refer to both the Euclidean norm and the "Fisher norm".)
- The paper briefly cites Kunstner et al. (2019), which is an explicit critique of the interpretation of Adam as NGD, but does not really engage with the arguments in that paper.
- Overall, the paper combines various components, that are somewhat independent of each other:
    a) introduce gradient clipping,
    b) apply momentum after preconditioning,
    c) apply preconditioning to the weight decay gradient.
It would be highly desirable to perform ablation studies to understand which of these changes actually matter and how they interact.
- The quality of the empirical evaluation is a bit lacking. No error bars are given. The hyperparameter tuning protocol is somewhat unclear, e.g., FAdam uses a different epsilon value and it is not stated how this value was obtained.

**Questions:**

- What is the exact argument for the "invariant natural gradient"?
- Kunstner et al. (2019) explicitly critique the interpretation of Adam as approximate NGD. What is your response to their arguments? (E.g., degeneracy of the empirical FIM for overparametrized models, sensitivity to model misspecification, no relationship between empirical and true FIM far from an optimum.)
- How were the $\epsilon$ values in the experiment chose?

---

### Official Review · Reviewer_2Fyw · 2024-11-10

**Soundness:** 3
**Presentation:** 3
**Contribution:** 2
**Rating:** 3
**Confidence:** 5

**Summary:**

This paper studies the connection between Adam optimizer and natural gradient descent by leveraging techniques from Riemannian geometry. Based on this, the authors propose a modified algorithm named Fisher Adam (FAdam). The convergence analysis of FAdam is provided and the algorithm is tested by large language model (LLM) experiments.

**Strengths:**

1. Understanding Adam is an important problem. Using fisher information and natural gradient descent to understand Adam is novel.

2. The presentation of this paper is good in general.

**Weaknesses:**

1. The theoretical analysis of FAdam is weak. It directly follows the paper (Defossez et al. 2020) and requires strong assumptions (e.g., $\beta_1$=0, bounded gradient). So it does not analyze the algorithm's momentum and is worse than the state-of-the-art analysis of Adam in the literature.

2. I do not find any rigorous presentation of Adam's flaws in the paper as claimed in the abstract by the authors. For example, the paper does not have any clear negative results of the vanilla Adam experimentally or theoretically. Therefore, the motivation of designing a new variant of Adam such as FAdam is unclear to me.

3. The description of the experiment is unclear. Lots of details are missing, such as training/test learning curve comparison, learning rate of the optimizer, batch size, and memory costs. Also, the experiment is only run once, and the algorithm's robustness is unclear.

**Questions:**

See weaknesses section.

---

### Note · Authors · 2024-11-27

I have read and agree with the venue's withdrawal policy on behalf of myself and my co-authors.